# Simultaneous enhanced efficiency and thermal stability in organic solar cells from a polymer acceptor additive

Wenyan Yang[1,5], Zhenghui Luo[2,5], Rui Sun[1], Jie Guo[1], Tao Wang[1], Yao Wu[1], Wei Wang[1], Jing Guo[1], Qiang Wu[1], Mumin Shi[1], Hongneng Li[1], Chuluo Yang [2] & Jie Min[1,3,4✉]

The thermal stability of organic solar cells is critical for practical applications of this emerging technology. Thus, effective approaches and strategies need to be found to alleviate their inherent thermal instability. Here, we show a polymer acceptor-doping general strategy and report a thermally stable bulk heterojunction photovoltaic system, which exhibits an improved power conversion efficiency of 15.10%. Supported by statistical analyses of device degradation data, and morphological characteristics and physical mechanisms study, this polymer-doping blend shows a longer lifetime, nearly keeping its efficiency ($t = 800$ h) under accelerated aging tests at 150 °C. Further analysis of the degradation behaviors indicates a bright future of this system in outer space applications. Notably, the use of polymer acceptor as a dual function additive in the other four photovoltaic systems was also confirmed, demonstrating the good generality of this polymer-doping strategy.

[1] The Institute for Advanced Studies, Wuhan University, 430072 Wuhan, China. [2] Hubei Key Lab on Organic and Polymeric Optoelectronic Materials, Department of Chemistry, Wuhan University, 430072 Wuhan, China. [3] Beijing National Laboratory for Molecular Sciences, 100190 Beijing, China. [4] Key Laboratory of Materials Processing and Mold, Zhengzhou University, Ministry of Education, 450002 Zhengzhou, China. [5] These authors contributed equally: Wenyan Yang, Zhenghui Luo. ✉email: min.jie@whu.edu.cn

Solution-processed organic solar cells (OSCs) have attracted considerable attention because of their advantages, such as lightweight and flexible substrates, and colorful and easy to mass produce[1–5]. Particularly, non-fullerene acceptors (NFAs) have been successfully applied to high-performance OSCs and have shown promising development in the past few years[6–8]. The power conversion efficiencies (PCEs) of over 16% have been reported for non-fullerene OSCs based on donor polymers[9–11]. Despite the great achievements in improving efficiency, commercialization of large-scale OSCs is limited due to their relatively low environmental stability[8,12–16]. Of note is that many environmental issues (e.g., heat, oxygen, light, and humidity) have been identified as main factors of device degradation[17]. These parameters should be properly addressed to meet the requirements of practical outdoor application of OSCs.

It is well established that device efficiency is strongly dependent on the bulk heterojunction (BHJ) nanoscale morphology, to effectively achieve optimized charge generation, carrier transport, and carrier collection[18–20]. However, the optimized blend morphology, known as a thermally activated metastable state, is generally sensitive to high-temperature aging[13,21], and easily moves toward a thermodynamic equilibrium state and thus forms a large phase separation, significantly reducing donor–acceptor (D–A) interfacial area and device efficiency[22,23]. For instance, storing the blend film of polymer donor PTB7-Th (poly[4,8-bis (5-(2-ethylhexyl)thiophen-2-yl)benzo[1,2-*b*;4,5-*b*′]dithiophene-2,6-diyl-alt-(4-(2-ethylhexyl)-3-fluorothieno[3,4*b*]thiophene-)-2-carboxylate-2-6-diyl)]) and fullerene acceptor PCBM (phenyl-C61-butyric acid methyl ester) in a 140 °C oven for 2 h induces the formation of numerous clusters, reducing the device efficiency by ~80%[24]. In addition to fullerene-based photovoltaic systems[12,24], many highly efficient NFA-based blends are also extremely sensitive to thermal stress[13,25,26]. These reported results suggest that the thermal instability of BHJ blends severely limit their outdoor applications, as these devices operating under standard conditions are obviously heated by prolonged illumination of sunlight[27,28], and the operating temperature for solar panels can be as high as 50–70 °C, even reaching 100 °C in some areas[29]. Although annealing effects have been extensively and deeply explored in OSCs[12,14–16,24,29,30], their device efficiencies have been temperature dependent, and cannot withstand temperatures well over 120 °C for extended periods of time.

Several effective approaches such as suitably reducing the crystallinity of photovoltaic polymers[31,32] or fullerenes[24,33], selecting suitable D/A pairs[34,35], incorporating a "molecular lock"[16,36], and cross-linking between D/A components in the active layers have been developed to solve thermal instability of BHJ blends[22,37,38]. Among them, blending host-active layers with additives emerges as a simple and general approach to achieve the optimal blend morphology, and also improves the electronic performance and environmental stability in OSCs[39–41]. The capability of packing motifs at elevated temperatures and preserving close intermolecular interactions is imperative but still lacking, especially for high-performance NFA systems[9,14,42,43]. We hypothesize that relevant conformational changes of interpenetrating networks between host donor and acceptor materials via introduction of semicrystalline conjugated polymers as additives can be suppressed at elevated temperatures.

Herein, we focus our endeavor to test this concept and solve the dilemma that BHJ photoactive layers are not resistant to high temperatures. In this work, an NFA molecule ((2,2′-((2Z,2′Z)-((12,13-bis(2-ethylhexyl)-3,9-diundecyl-12,13dihydro-[1,2,5]thiadiazolo[3,4-*e*]thieno[2″,3‴:4′,5′]thieno[2′,3′:4,5]pyrrolo[3,2-*g*]thieno[2′,3′:4,5]thieno[3,2-b]indole-2,10-diyl)bis(methanylylidene))bis(2-chloro-6-oxo-5,6-dihydro-4*H*-cyclopenta[*b*]thiophene-4-ylidene)dimalononitrile)) (BTTT-2Cl, Fig. 1a) is developed. Optimized

BTTT-2Cl-based devices show a PCE of up to 13.83% by incorporating the commercially available polymer donor PM6 (as illustrated in Fig. 1a)[44]. Adding a small amount (1 wt%) of semicrystalline photovoltaic polymer acceptor PZ1[45] into the PM6:BTTT-2Cl host blend improved the crystalline morphology of active layer and enhanced its charge transport property, achieving a PCE of 15.10%. In parallel, the excellent film quality effectively suppressed the thermally driven phase separation at elevated temperatures, and was thermally stable at 150 °C for over 800 h in the nitrogen-filled glovebox. Furthermore, this PZ1-doping blend also exhibits robust morphology against specific thermal-cycling stress conditions. Besides, the use of PZ1 employed as the dual function additive for improving photovoltaic efficiency and thermal stability was also confirmed in other four commonly used photovoltaic systems, suggesting that the polymer acceptor PZ1 is a universal solid additive for highly thermal-stable OSC applications.

## Results

**Material properties and film morphology.** The chemical structures of the synthesized NFA material, BTTT-2Cl, and the polymer donor PM6 and acceptor-additive PZ1 are shown in Fig. 1a. The synthetic routes and the theoretically calculated molecular energy levels of BTTT-2Cl are provided in Supplementary Fig. 1. The synthetic details (Supplementary Figs. 1 and 2 for the nuclear magnetic resonance (NMR) spectra, and Supplementary Fig. 3 for the mass spectrum) are described in Supplementary Note 1. Compared to the solution absorption spectra of BTTT-2Cl, an obvious bathochromic shift of ~90 nm in its thin film (Supplementary Fig. 5) indicates high molecular ordering[46]. The thermal properties of these three materials, including PM6, BTTT-2Cl, and PZ1, were investigated by differential scanning calorimetry. The corresponding results are provided in Supplementary Fig. 6. Only BTTT-2Cl shows single, sharp, and strong endothermic peak at the first heating (Supplementary Fig. 6a), which indicates its crystalline nature. As shown in Supplementary Fig. 6a, PM6 has slightly sharper endothermic peaks at low temperature and with small value of melting enthalpy. It indicates that PM6 possesses slightly higher crystallinity as compared to PZ1 (Supplementary Fig. 6c), which is identical to the previous results[11,45]. The ultraviolet–visible absorption spectra of the pristine films and the PM6:BTTT-2Cl blend films without and with 1 wt% PZ1, as shown in Fig. 1b, were explored. The absorption maxima of PM6:BTTT-2Cl blend shows a blue-shift compared to that of the BTTT-2Cl pristine film. Obviously, the addition of 1 wt% PZ1 into the host blend presents an obvious increase in the 0–0 vibronic peak intensity with a slight red-shift, indicating the increasing molecular ordering and π–π interactions in the nanoscale domains[47]. The highest occupied molecular orbital (HOMO) and lowest unoccupied molecular orbital (LUMO) energy levels of BTTT-2Cl, probed by cyclic voltammetry (CV) (Supplementary Fig. 7 and Supplementary Table 1) are −5.61 and −3.98 eV, respectively. The energy levels of PM6 and PZ1 are presented in Fig. 1c[48]. It indicates that the PM6:BTTT-2Cl combination is comfortable for exciton dissociation in BHJ blend, as demonstrated by the photoluminescence (PL) measurements (Supplementary Fig. 8). Of note is that the PL intensity of PM6:BTTT-2Cl blend increases with the addition of 1 wt% PZ1, indicating that the interface area between PM6 and BTTT-2Cl decreases[49].

This changed blend morphology was observed by atomic force microscopy (AFM) images (Fig. 1d) and two-dimensional (2D) grazing incidence wide-angle X-ray scattering (2D-GIWAXS) measurements (Fig. 1d), as well as their corresponding intensity profiles in the in-plane (IP) and out-of-plane (OOP) directions (Supplementary Fig. 9). The surface morphologies of AFM

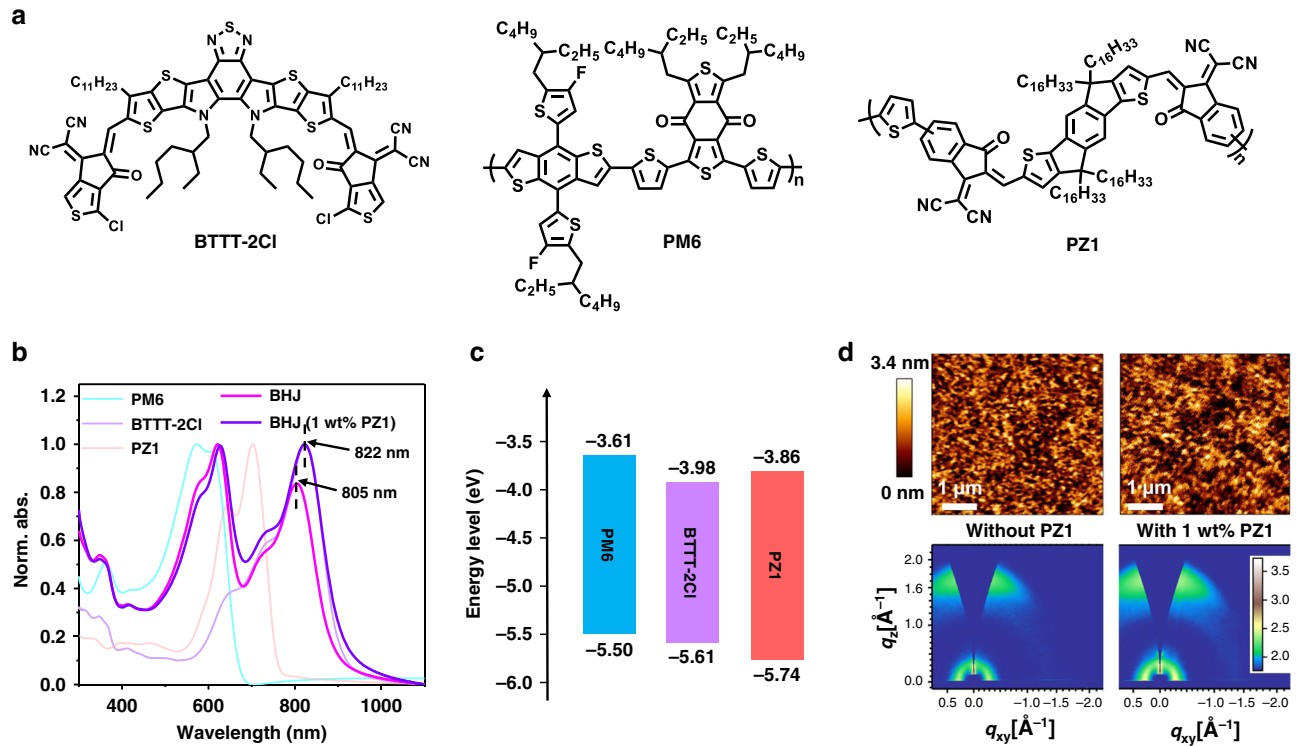

**Fig. 1 Chemical structures, material characterizations, and blend morphologies. a** Molecular structures of small-molecule acceptor BTTT-2Cl, polymer donor PM6, and acceptor additive PZ1. **b** Normalized UV–vis absorption spectra of the neat films and the PM6:BTTT-2Cl blends without and with dynamics wt% PZ1. **c** Energy level diagram of PM6, BTTT-2Cl, and PZ1. **d** Top: Tapping-mode AFM surface scans (size: 5 × 5 μm²) of the PM6:BTTT-2Cl blends without (a root mean square (RMS) roughness of 0.813 nm) and with 1 wt% PZ1 (an RMS roughness of 0.628 nm), the 3.4-nm scale bar applies to AFM images; bottom: 2D-GIWAXS patterns of the corresponding thin films acquired at the critical incident angle of 0.13°. The intensities are shown on a logarithmic scale, as indicated by the color bar.

images preliminarily demonstrated that the 1 wt% PZ1-doped blend shows larger interconnected domains and regions as compared to the host blend. The neat BTTT-2Cl film (Supplementary Fig. 9a) shows a strong π–π stacking peak in OOP direction at $q = 1.71\,\text{Å}^{-1}$ ($d \approx 3.67\,\text{Å}$). There exists one diffraction peak in the IP direction at $q = 1.47\,\text{Å}^{-1}$ ($d \approx 4.27\,\text{Å}$), suggesting the co-existence of two distinct structure orders[50]. The optimized PM6:BTTT-2Cl blend film exhibits a strong diffraction peak in the OOP direction at $q = 1.70\,\text{Å}^{-1}$ ($d \approx 3.69\,\text{Å}$) associated with the π–π stacking of BTTT-2Cl. Furthermore, the PZ1-doped host blend shows a stronger crystallization tendency confirmed by 2D-GIWAXS profiles. We calculated the crystal size by taking the full-width half-maximum (FWHM, using the Scherrer's equation) of the OOP (0 1 0) peak[51]. By adding 1 wt% PZ1, the crystal coherence length increased from 19.94 to 21.68 nm, consistent with the corresponding AFM images as presented in Fig. 1d. Overall, the morphological characteristics combined with the photophysical and chemical properties of PZ1 can give a clear conclusion that a small amount of PZ1 work as a solid additive in this PM6:BTTT-2Cl system.

**Photovoltaic performance of OSCs and their thermal stability.** The photovoltaic devices were fabricated with a conventional architecture consisting of indium tin oxide (ITO)/poly(3,4-ethylenedioxythiophene):polystyrenesulfonate (PEDOT:PSS)/photoactive layer/perylene diimide functionalized with amino N-oxide (PDINO)/Al. The details of device fabrication are described in the Methods section. In addition, the detailed photovoltaic performances are shown in Supplementary Figs. 10–12 for the

investigated devices without and with PZ1 additives, and the detailed photovoltaic parameters are listed in Supplementary Tables 2–4. The current density–voltage (J–V) curves of the corresponding best-performing devices are plotted in Fig. 2a. On the basis of a 1:1.2 blend ratio, the optimized PM6:BTTT-2Cl device exhibits a PCE of 13.80% ($V_{oc}$ (open-circuit voltage) = 0.896 V, $J_{sc}$ (short-circuit current density) = 23.80 mA cm⁻², FF (fill factor) = 65.19%). Here we introduced PZ1 acceptor as solid additives into the host blend for improving its device performance (Supplementary Figs. 11–13 and Supplementary Table 3). The peak efficiency, achieved at 1 wt% PZ1 yielded a high PCE of 15.10% ($V_{oc}$ = 0.904 V, $J_{sc}$ = 24.58 mA cm⁻², FF = 67.90%). Of note is that the slight voltage shift ($\Delta V_{oc}$ = 0.008 V) probably resulted from the high LUMO level of polymer PZ1 and the less charge carrier recombination loss, which will be mentioned below. The calculated $J_{sc}$ values obtained from the external quantum efficiency (EQE) spectra (Fig. 2b) were consistent with the measured $J_{sc}$ values, as provided in Table 1. Besides, the J–V curve of the PM6:PZ1 device with a weight ratio of 1.75:1 is shown in Supplementary Fig. 14. The best efficiency showed a PCE of 8.27% ($V_{oc}$ = 0.94 V, $J_{sc}$ = 15.34 mA cm⁻², FF = 57.65%). It is lower than the published value (11.2%) reported in ref. [48], probably resulting from the different molecular weights of PM6 and PZ1.

We further employed space charge-limited current (SCLC) to determine the hole and electron mobilities in relevant devices (Fig. 2c and Table 1). The average hole-only mobility of PZ1-doped devices ($\mu_h = 1.38 \times 10^{-3}$ cm² V⁻¹ s⁻¹, Supplementary Fig. 15) is higher than that of undoped devices ($\mu_h = 6.88 \times 10^{-4}$ cm² V⁻¹ s⁻¹, Supplementary Fig. 15). In addition, the

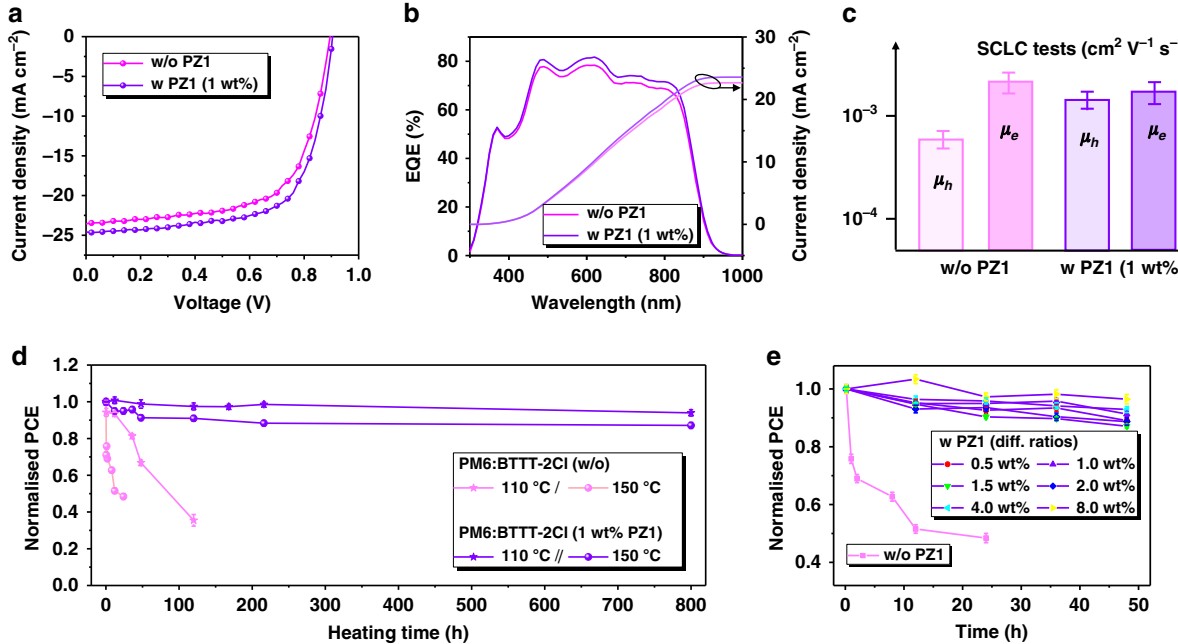

**Fig. 2 Photovoltaic performance and thermal stability of the blends. a** $J$–$V$ curves of the champion devices fabricated without and with 1 wt% PZ1. **b** EQE spectra and integrated current densities from the EQE spectra. **c** Hole- and electron-only mobilities of relevant devices without and with 1 wt% PZ1. **d** Normalized photovoltaic performance of the investigated PM6:BDTTT-2Cl blends without and with 1 wt% PZ1 as a function of annealing time at 150 °C and 110 °C, respectively. **e** Normalized PCE of corresponding blends with different PZ1 concentrations as a function of annealing time at 150 °C. Source data are provided as a Source Data file.

**Table 1 Detailed photovoltaic parameters and carrier mobilities of the relevant blends.**

| PZ1 concentration (wt%) | $V_{oc}$ (V) | $J_{sc}$ (mA cm$^{-2}$) | Calculated $J_{sc}$ (mA cm$^{-2}$) | FF (%) | PCE (%) | | Mobility (cm$^2$ V$^{-1}$ s$^{-1}$) | |
|---|---|---|---|---|---|---|---|---|
| | | | | | Average | Best | $\mu_h$ | $\mu_e$ |
| 0 | 0.896 | 23.80 | 22.66 | 65.19 | 13.30 ± 0.54 | 13.80 | $6.88 \times 10^{-4}$ | $2.25 \times 10^{-3}$ |
| 1 | 0.904 | 24.58 | 23.57 | 67.90 | 14.69 ± 0.45 | 15.10 | $1.38 \times 10^{-3}$ | $2.07 \times 10^{-3}$ |

average electron-only mobility ($\mu_e$) values of the relevant devices without and with 1 wt% PZ1 are $2.25 \times 10^{-3}$ and $2.07 \times 10^{-3}$ cm$^2$ V$^{-1}$ s$^{-1}$ (Supplementary Fig. 16), respectively. The comparable $\mu_e$ values of the PZ1-doped and undoped blends are probably contributed to the zinc oxide (ZnO) morphology as well as its lower surface energy as compared to the PEDOT:PSS layer, which can affect the film formation of the active layers[51]. Nevertheless, combining with the more balanced transport of holes and electrons in the PZ1-doped devices, the SCLC data illustrate that charge carriers in the 1 wt% PZ1-doped blend can be transmitted more efficiently, which is also evidenced by the morphological characterizations (Fig. 1d). In addition, the relative recombination loss as a function of light intensity was investigated in these devices without and with 1 wt% PZ1, as plotted in Supplementary Fig. 17. It was found that adding a small amount of PZ1 into the host devices did not exhibit obvious bimolecular recombination (Supplementary Fig. 17a). In contrast, the PZ1-doped OSCs showed less trap-assisted recombination with reduced slop of 1.36 kT/q (versus 1.44 kT/q for the host devices, Supplementary Fig. 17b). The detailed analyses are provided in Supplementary Note 2. Overall, the doping of PZ1 polymer not only improves the blend morphology with suitable phase separation and large pure domains but also can reduce the carrier recombination loss and enhance the carrier transport properties in the doping blend.

Importantly, we conducted a thermal stress stability test for the PM6:BTTT-2Cl blends without and with 1 wt% PZ1 at 110 °C and 150 °C, respectively, in a nitrogen atmosphere. The average

efficiency of the undoped blend degraded gradually from 13.30% to 6.38% and then 4.66% after constant heating at 150 °C for 24 h and 110 °C for 120 h, respectively, which represent 52% and 65% loss in efficiencies. In contrast with the host blends under the same conditions, the blends with 1 wt% PZ1 showed excellent accelerated thermal stability depicting 5% and 12% efficiency losses after heating at 110 °C and 150 °C for 800 h, as exhibited in Fig. 2d. Moreover, the devices with added PZ1 weight ratios in the range 0.5–8.0% retained 88–97% of their PCEs after 48 h, as presented in Fig. 2e. Unsurprisingly, as shown in Supplementary Fig. 18, the above-mentioned PM6:PZ1 all-polymer photovoltaic system also retained ~90% of its PCEs after 48 h, indicating the super thermal stability at a high temperature. These results indicate that PZ1 as a solid additive has played a significant role in mitigating morphological degradation, which is mainly due to the thermally driven phase separation and BTTT-2Cl aggregations as discussed below.

**Morphology and physical mechanism analysis under thermal stress.** As discussed above, the addition of PZ1 results in a favorable blend film with more obvious molecular ordering of BTTT-2Cl in comparison with the blend film without PZ1 (Fig. 1d). As shown in Fig. 3a, molecular dynamics modeling was performed to investigate the influence of the π–π stacking confinement on the molecular dihedral distributions. The resulting dihedral distributions were compared to characterize the BTTT-

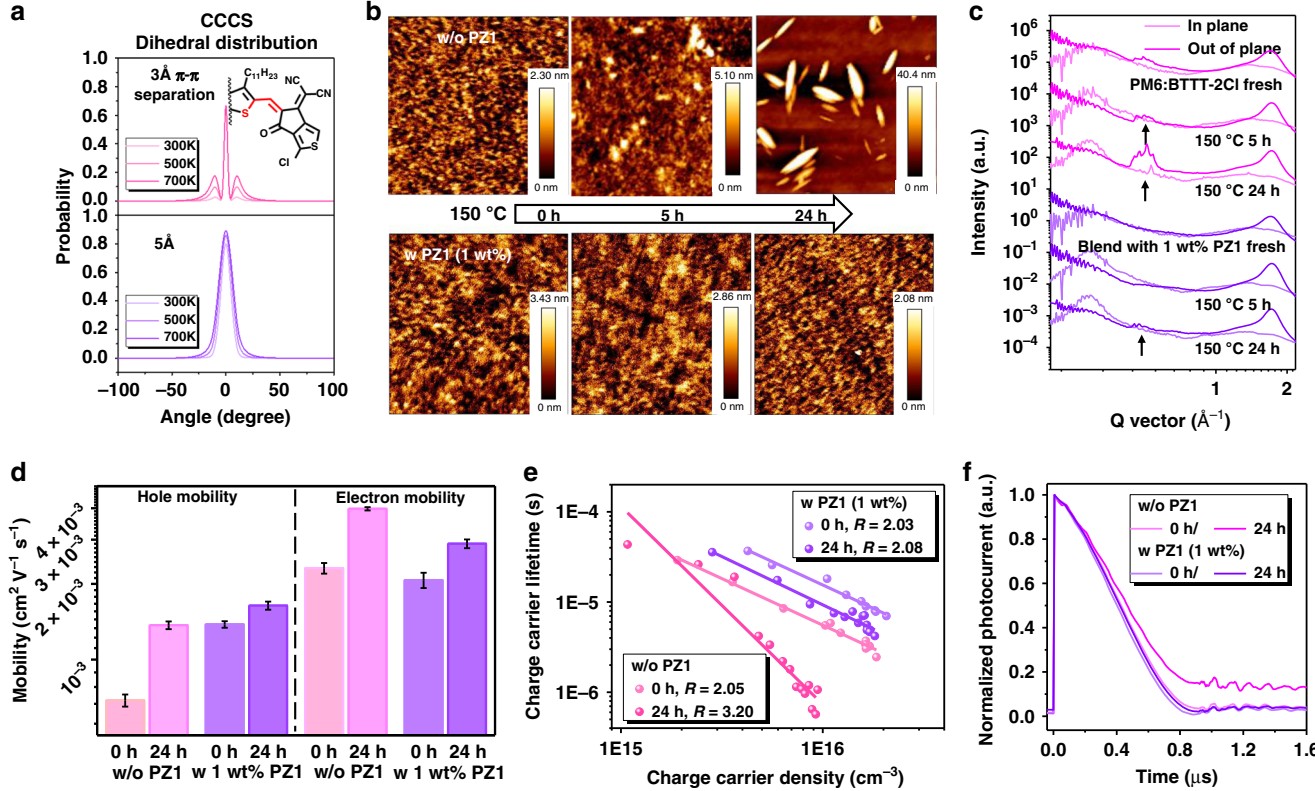

**Fig. 3 Morphological characteristics and physical mechanisms of the blends. a** Simulated dihedral distributions when π–π confinements are confined to 3 and 5 Å, respectively. The distribution broadens indicate the chains twisting at different temperatures. **b** Surface topographic AFM images (size: 5 μm × 5 μm) of the undoped and PZ1-doped films under thermal stress. **c** The one-dimensional GIWAXS line curves of the corresponding blends with respect to the OOP and IP directions. **d** The hole and electron mobilities of the corresponding films as a function of heating time at 150 °C. **e** Charge carrier lifetime obtained from TPV measurements as a function of charge carrier density ($n$) obtained from CE measurements under open-circuit conditions. **f** Normalized transient photocurrent (TPC) data for the undoped and PZ1-doped devices based on the relevant blends under thermal stress. Source data are provided as a Source Data file.

2Cl reorganization dynamics. It is found that the CCCS dihedral angle, corresponding to the D–A conformations, remained sharply peaked at π–π confinements of 3.0 Å. At π–π confinements of 5.0 Å, the CCCS dihedral distribution in contrast is broadened at the corresponding temperatures. These results indicate that the π–π confinement exhibited by the obvious phase separation in PZ1-doped blends played a vital role in enabling temperature-insensitive molecule migration. It suggests that this PZ1-doping approach can be general to other photovoltaic systems, which will be discussed in the last section.

To understand the thermal stability of the PZ1-doped BHJ blend versus host blend, we further conducted AFM and GIWAXS measurements to investigate the morphological change of the relevant BHJ blends. The surface images before and after the heating at 150 °C for 5 and 24 h are shown in Fig. 3b. The AFM images of the PM6:BTTT-2Cl blend as a function of annealing time at 150 °C are shown in Supplementary Fig. 19. The homogeneity of the PM6:BTTT-2Cl BHJ blend was altered subsequent to the heating. Many long string-like BTTT-2Cl crystals were observed in the aged blend. In contrast, the PZ1-incorporated blend morphology was not significantly affected by heating. These images provide a direct evidence that PZ1 as a solid additive is very efficient in limiting the thermally induced BTTT-2Cl migration and aggregation in the blend and also reducing its phase segregation. Incorporating polymer acceptor PZ1 with long alkyl side chains into the PM6:BTTT-2Cl blend can form a strong framework and also inhibit BTTT-2Cl aggregation and crystallization. In addition, 2D-GIWAXS was also conducted to investigate the thermally induced crystallization

of the BTTT-2Cl molecules in the blends (Fig. 3c, Supplementary Fig. 20, and Supplementary Table 5). The (0 0 1) diffraction peaks, which can be defined as parallel orientation to the substrate, exhibited prominent intensity in the PM6:BTTT-2Cl blend under prolonged thermal stress. In contrast, when 1 wt% PZ1 was provided into the control BHJ blend, the intensity growth of the GIWAXS profiles can be suppressed significantly, resulting in the low device efficiency losses at high temperatures.

Figure 3d exhibits the hole and electron mobilities of the relevant blends before and after heating at a temperature of 150 °C for 24 h. As presented in Supplementary Figs. 15 and 16, the hole- and electron-only mobilities of the relevant BHJ blends were significantly increased after thermal annealing for 24 h, with the host blend gaining more carrier transport properties than the PZ1-doped blend. In addition to the investigation of charge transport properties in these two blends under prolonged thermal stress, we also directly determined their carrier recombination mechanisms. We combined the charge extraction (CE) and transient photovoltage (TPV) techniques to yield the charge carrier lifetime ($\tau$) as a function of charge carrier density ($n$) under $V_{oc}$ conditions (Supplementary Figs. 21 and 22)[52]. Here, a non-geminate recombination order $R$ ($R = \lambda + 1$) was calculated via the equation $\tau = \tau_0(n_0/n)^\lambda$, where $\lambda$ is the so-called recombination exponent, and $\tau_0$ and $n_0$ are constants achieved from the fitting data. As exhibited in Fig. 3e, a slightly higher recombination order value ($R = 2.05$) for the undoped BHJ solar cell compared to the PZ1-doped device ($R = 2.03$) was observed. After 24 h heating, it was found that the host blend exhibited an increased recombination with an $R$ of 3.20. In

contrast, the changes of the recombination order with an $R$ of 2.08 in the PZ1-doped device are not obvious.

The dwell time of charges in the relevant blends prior to extraction at the electrodes can be measured by transient photocurrent technique[53]. The reduced recombination order value demonstrated the slightly $V_{oc}$ improvement in the PZ1-doped device. Figure 3f shows the CE properties of the undoped and PZ1-doped devices at $J_{sc}$ condition, where the internal field equals the built-in potential ($V_{bi}$). The extraction time of the undoped solar cell was calculated to be $\tau = 0.36\,\mu s$ for 0 h and $\tau = 0.47\,\mu s$ for 24 h heating. In the case of the PZ1-doped devices before and after thermal stress, they showed very short lifetimes of 0.34 μs for 0 h and 0.36 μs for 24 h, respectively. The more stable recombination order value and extraction lifetime suggest that the more efficient CE ability in the PZ1-doped devices as compared to that of the host devices based on the corresponding blends heating at 150 °C for 24 h. These results are also consistent with the difference in blend morphology observed in the undoped and PZ1-doped blends (Fig. 3b, c). Briefly, the results of the molecular dynamics simulations and blend morphology investigations coupled with the analysis of carrier transport and recombination mechanisms finally underpin the photovoltaic performance deterioration of the corresponding blends under prolonged thermal stress and give detailed insight into the PZ1-doping strategy responsible for optimizing and fixing the morphology of photoactive layer.

**Simulation of OSCs' application in outer space**. The above-discussed thermal stability studies are focused on the long-term thermal degradation of the PZ1-doped PM6:BTTT-2Cl blend, indicating that it has the capability to successfully work after experiencing the extreme temperature at 150 °C. Since the relevant stability of photovoltaic systems under rapidly changing wide temperature range, thermal-cycling conditions is lacking, in this work we further evaluated the prolonged stability of the PZ1-doped blends conducted at the simulated space environments, including Moon, Low Earth Orbital Satellite, and Mars. The testing profiles of the thermal-cycling tests were designed and defined as depicted in Fig. 4a, c to simulate the moon, Low Earth Orbital Satellite, and Mars environments, respectively, and verify the thermal-cycling stability of the PZ1-doped PM6:BTTT-2Cl blend. Details of thermal-cycling test profiles of the blends under the different spaces are listed in Supplementary Table 6, and the experimental results are obtained as also shown in Fig. 4a, c. Of note is that the thermal stability issues are significantly investigated here. Thus, the irradiation stability of relevant active layers was not considered in the simulated environment of outer space applications. Under the simulated stress conditions based on Moon[54], Low Earth Orbital Satellite, and Mars, respectively, over 90% of the original device efficiency can be retained after several thermal cycles. Obviously, the degradation behaviors of the PZ1-doped blends, subjected to thermal-cycling stress conditions in which the temperature changes in a short period of time, are not obvious. Of note is that this result is relevant not only to the outer space applications but also to the real-world outdoor applications. Supplementary Figure 23 shows the durability of the stored OSCs under thermal-cycling stress between −20 °C and 80 °C for 600 h. The investigated devices exhibited great long-term stability under thermal-cycling stress even when metal electrodes and rigid glass substrates were employed in this device structure. Overall, these findings suggest that the PZ1-doped PM6:BTTT-2Cl photovoltaic system has the potential to be applied successfully in extreme environmental conditions with alternating temperatures, similar to the space applications. Based on this point, we further put forward the summary PCE estimates under different simulated

conditions, which are derived as a function of the optical bandgap of photoactive layer, as provided in Fig. 4d. We found PCE to be >20%, which was calculated using the AM 1.5 spectrum when we simultaneously used an FF = 80% and a constant EQE = 85%. Due to the higher solar light intensity on the surface of the Moon and Satellite (AM 0 spectrum), the device efficiency can be further improved to over 24% for $E_g$ 1.28 to 1.59 eV (780–969 nm). Since the solar irradiance is 590 W m$^{-2}$ at local noon on Mars, which is lower than that of the Earth's surface (1000 W m$^{-2}$), the maximum evaluated PCE is ~12%.

**Universality of the PZ1-doping strategy**. Apart from the fact that PZ1 additive can efficiently enhance the thermal stability of the PM6:BDTTT-2Cl blend, here we also explored the thermal stability of PZ1 doped with four other photovoltaic systems, including PM6:Y6[46], J71:ITIC[55], PTB7-Th:PC$_{70}$BM[56], and BDT-3T-R:PC$_{70}$BM[57] (Fig. 5a), for demonstrating the generality of our doping strategy. The detailed photovoltaic performance of the investigated devices with different contents of PZ1 additives are shown in Supplementary Figs. 24–27, and the detailed photovoltaic parameters are listed in Supplementary Tables 7–10. In addition, after heating at 150 °C for 48 h (Fig. 5b), the PCE of PM6:Y6 blends without PZ1 decays from 15.33% to 11.40% (Supplementary Fig. 28 and Supplementary Table 11); the PCE of J71:ITIC blends without PZ1 decays from 10.06% to 8.00% (Supplementary Fig. 29 and Supplementary Table 12); the PCE of PTB7-Th:PC$_{70}$BM blends without PZ1 decays from 8.39% to 2.65% (Supplementary Fig. 30 and Supplementary Table 13); the PCE of BDT-3T-R:PC$_{70}$BM blends without PZ1 decays from 6.68% to 0.56% (Supplementary Fig. 31 and Supplementary Table 14), preserving 74%, 79%, 31%, and 8.4% of their original average values, respectively. The addition of PZ1 improves the thermal stability of these non-fullerene and fullerene-based blends, as presented in Fig. 5b. After heating at 150 °C for 48 h, the PCE of the 1 wt% PZ1-doped PM6:Y6 blends decays from 15.74 to 13.86% (Supplementary Fig. 32 and Supplementary Table 15); the PCE of the 1 wt% PZ1-doped J71:ITIC blends decays from 10.76 to 9.97% (Supplementary Fig. 33 and Supplementary Table 16); the PCE of the 1 wt% PZ1-doped PTB7-Th:PC$_{70}$BM blends decays from 8.79 to 5.10% (Supplementary Fig. 34 and Supplementary Table 17); the PCE of the 1 wt% PZ1-doped BDT-3T-R:PC$_{70}$BM blends decays from 7.04 to 1.82% (Supplementary Fig. 35 and Supplementary Table 18). The PCEs of these four systems upon high temperature annealing for 48 h preserved 88%, 92%, 58%, and 26% of their original average values, respectively. Besides, the corresponding blends with the addition of different contents of PZ1 (1.5, 2.0, and 4.0%) also show the improved thermal stability properties (Fig. 5b). Of note is that the PCEs of these four photovoltaic systems are also improved by adding 1 wt% PZ1 into the host blends, as exhibited in Supplementary Figs. 24–27, and summarized in Supplementary Tables 7–10. These results further demonstrated the good generality of the PZ1-doping strategy.

**Discussion**

In summary, we designed and synthesized an NFA, BTTT-2Cl. Using polymer PM6 as the donor, we systematically evaluated the photovoltaic properties of PM6:BTTT-2Cl solar cells with a PCE of 13.80%. We further introduced a polymer acceptor PZ1 employed as a dual functional solid additive. The addition of 1 wt % PZ1 not only enhances π–π stacking and aggregation of BTTT-2Cl acceptors promoting device efficiency up to 15.10% but also dramatically enhances its thermal stability at high temperatures. The morphological and physical characterizations demonstrated that PZ1 can significantly suppress the phase separation, fix the blend microstructure, and slow down the trap generation of the PZ1-doped blend. This PZ1-doped blend also exhibits great

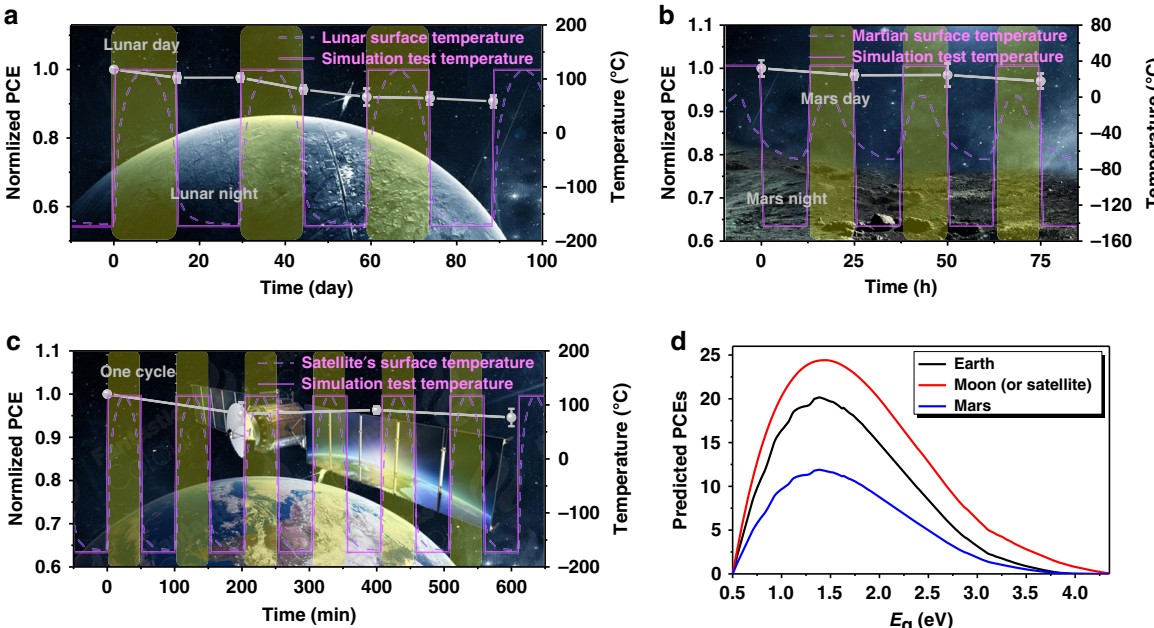

**Fig. 4 Thermal cycle stability and theoretical efficiency of photovoltaic devices.** Thermal cycle stability of the PZ1-doped blends (**a**) under the simulated thermal stress of Moon environment, **b** under the simulated thermal stress of Low Earth Orbital Satellite, and **c** under the simulated thermal stress of Mars environment. **d** Theoretical efficiency of non-fullerene-based solar cells with $E_g - eV_{oc} = 0.5$ eV versus the lowest $E_g$ value of the blend films, which was calculated using the 100% AM 1.5 spectrum for the Earth and 59% AM 1.5 spectrum for the Mars[58], and the AM 0 spectrum for the Moon and Satellite. The $E_g$ value is determined by the broader optical absorption spectra of donor or acceptor in the film. For the purpose of calculation, an FF of 80% and an EQE of 85% were assumed. Source data are provided as a Source Data file.

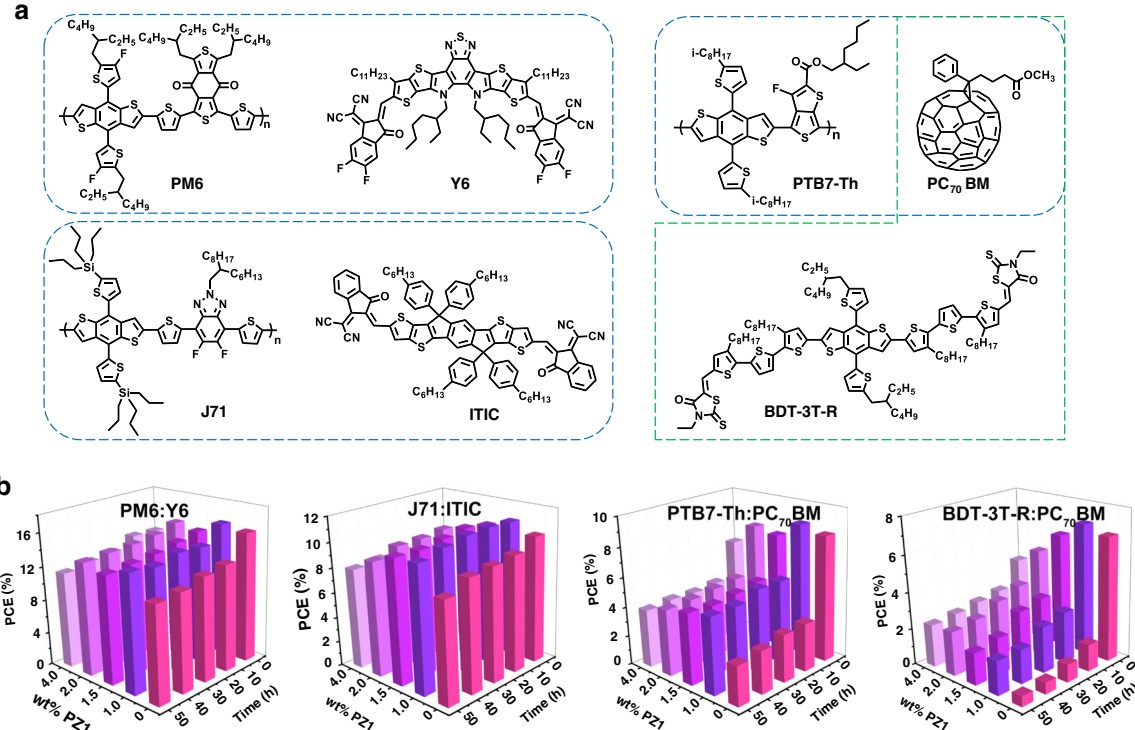

**Fig. 5 Chemical structures and thermal stability of photovoltaic systems. a** Molecular structures of four photovoltaic systems, including PM6:Y6, J71:ITIC, PTB7-Th:PC₇₀BM, and BDT-3T-R:PC₇₀BM. Normalized photovoltaic performance of investigated (**b**) PM6:Y6, J71:ITIC, PTB7-Th:PC₇₀BM, and BDT-3T-R:PC₇₀BM blends doped by PZ1 additives as a function of annealing time at 150 °C.

durability under thermal-cycling stress, indicating a bright future in outer space applications. Importantly, based on the analysis of the other four photovoltaic systems, our results demonstrate that the use of polymer acceptor PZ1 as a dual function additive is a simple and general doping strategy for simultaneously improving photovoltaic performance and thermal stability of OSCs. Hence, this work provides a polymer-doping strategy to develop efficient and thermally stable organic photovoltaic systems for their perspective of industrial applications.

## Methods

**Materials**. PTB7-Th, ITIC, and PC70BM were provided by Solarmer Materials Inc. PM6, J71, BDT-3T-R, and Y6 were synthesized according to the procedures in the literatures[1–4]; the synthetic route of BTTT-2Cl was shown in Supplementary Fig. 1a.

**Material characterization**. [1]H NMR spectra were measured on a Bruker DMX-400 spectrometer. The chemical shifts were reported as $\delta$ value (p.p.m.), which is relative to an internal tetramethylsilane standard. A Perkin-Elmer Lambda-35 absorption spectrometer was used to record the absorption profiles. The CV, which was conducted on a CS350H electrochemical workstation with a three-electrode system in a 0.1 M Bu$_4$NPF$_6$ acetonitrile solution at a scan rate of 20 mV s$^{-1}$, was used to study the electrochemcial properties. The LUMO/HOMO energy levels ($E_{LUMO}/E_{HOMO}$) can be calculated from onset reduction/oxidation potentials ($\phi_{red}/\phi_{ox}$) in the CVs according to the equations of $E_{LUMO}/E_{HOMO} = -e$ ($\phi_{red}/\phi_{ox} + 4.8 - \phi_{Fc/Fc^+}$) (eV)[4], where $\phi_{Fc/Fc^+}$ is the redox potential of ferrocene/ferrocenium (Fc/Fc$^+$) couple in the electrochemical measurement system. The relevant energy level of Fc/Fc$^+$ was taken as 4.8 eV below vacuum. In addition, DSC scans were obtained with Mettler Toledo DSC30 system with 10 °C min$^{-1}$ heating/cooling rate in temperature range of 20–290 °C.

**Fabrication and characterization of the OSCs**. All the solar cells were fabricated in the conventional architecture. For the device fabrication, the ITO substrates were cleaned in toluene, water, acetone, and isopropyl alcohol. After drying, the ITO substrates were spin coated with 40 nm PEDOT:PSS (HC Starck, PEDOT PH-4083). The photovoltaic layer was spin coated in a nitrogen glovebox from a mixed solution of PM6:BTTT-2Cl (1:1.2, wt%) with 15 mg mL$^{-1}$ in chloroform without and with the different concentration of PZ1 additives. In addition, the PM6:PZ1 blend was spin coated from a solution of PM6:PZ1 (1.75:1, wt%) with 20 mg mL$^{-1}$ in chloroform. Besides, the other four photovoltaic layers, consisting of four different photovoltaic systems as shown in Fig. 5, were dissolved in chloroform (PM6:Y6, J71:ITIC, and BDT-3T-R:PC$_{70}$BM) or ODCB (PTB7-Th:PC$_{70}$BM) with various weight ratios, and spin coated on top of the PEDOT:PSS layer. A PDINO layer via a solution concentration of 1.0 mg mL$^{-1}$ was deposited at the top of the active layer. Finally, the top aluminum electrode of 100 nm thickness was evaporated in vacuum onto the cathode buffer layer at a pressure of $5 \times 10^{-6}$ mbar. The active area of the solar cells was 4 mm$^2$. The device efficiencies were measured by a Keithley 2400 source meter unit under AM 1.5G (100 mW cm$^{-2}$) irradiation from a solar simulator (Enlitech model SS-F5-3A). A monocrystalline silicon reference cell with KG5 filter was used to determine the illumination intensity of solar simulator. The EQE was measured by a Solar Cell Spectral Response Measurement System QE-R3011 (Enli Technology Co., Ltd.).

**AFM measurements**. The surface morphologies of AFM images relative to the corresponding BHJ blends were measured by a Nano Wizard 4 AFM (JPK Inc. Germany) in Qi mode.

**SCLC measurements**. The structure of hole-only devices was Glass/ITO/PEDOT:PSS/Active layer/MoO$_3$/Ag (100 nm). In addition, the structure of electron-only mobility was Glass/ITO/ZnO/Active layer/Ca (15 nm)/Ag (80 nm). The dark current–voltage characteristics of single carrier devices were measured and analyzed in the SCLC regime following refs. [21,49].

**GIWAXS measurements**. The 2D-GIWAXS measurements were conducted on a Xenocs-SAXS/WAXS system with X-ray wavelength of 1.5418 Å. The shallow incident angle scattering was collected at 0.02°. All samples are prepared by spin-coating chloroform solutions on PEDOT:PSS-based glass substrates.

**Photo-induced charge carrier extraction by linearly increasing the voltage measurements**. The investigated solar cells are illuminated with a 405-nm laser diode. A fast electrical switch was applied to isolate the device in order to prevent carrier extraction or sweep out. After the variable delay time, the switch connected the device to a function generator. In addition, current transients are recorded across the internal 50 Ω resistor of our oscilloscope. To investigate the carrier mobility in the corresponding devices, the photo-CELIV (charge carrier extraction by linearly increasing the voltage) curves were conducted using different experimental conditions, differing in delay time and applied voltage.

**TPV and CE measurements**. For TPV measurements, a small optical perturbation was applied using a 405-nm laser diode, which was adjusted in light intensity to produce a voltage perturbation of $\Delta V_o < 10$ mV $\ll V_{oc}$. The corresponding devices were illuminated with a white-light LED (light-emitting diode) at different light intensities. The amount of charges generated by the pulse was obtained by integrating a photocurrent measurement. For CE measurements, the corresponding devices were held at a specified voltage in the dark or under different light intensities. The initial device voltage was set by a Keithley 2440 source-measurement unit. In addition, a fast analog switch from Texas Instruments (TS5A23159) is used to perform the switching from the specified voltage to short-circuit conditions. It provides a low on-state resistance (1 Ω), a quick switching time (50 ns), a very low charge injection ($\ll 10^{15}$ cm$^2$ V$^{-1}$ s$^{-1}$), and high off-state resistance ($>1$ MΩ).

**PL measurements**. The PL data and emission of relevant films were collected using a Zolix Flex One spectrometer. The PL excitation wavelength was set to 639 nm.

**Reporting summary**. Further information on research design is available in the Nature Research Reporting Summary linked to this article.

## Data availability

The relevant data are available from the authors upon reasonable request. In addition, the source data underlying Figs. 2c, 3d, 4a–c and Supplementary Figs. 12, 19, as well as Tables 1, 2, 3, 4, 7, 8, 9, and 10 are provided as a Source Data file.

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

## Acknowledgements

This work was financially supported by the National Natural Science Foundation of China (NSFC) (Grant Nos. 21702154 and 51773157). We also thank the support of the opening project of Key Laboratory of Materials Processing and Mold and Beijing National Laboratory for Molecular Sciences (BNLMS201905).

## Author contributions

W.Y.Y. and J.M. conceived and developed the ideas. W.Y.Y. designed the experiments and performed device fabrication. Z.H.L. and C.L.Y. synthesized the BTTT-2Cl. W.Y.Y. and R.S. performed TMU measurements. Jie G. carried out molecular dynamics simulation. T.W. and H.N.L. synthesized the PM6 and J71 materials. Y.W. synthesized the BDT-3T-R material. W.W. carried out CV test and analysis. Jing G. and M.M.S. performed PL measurement. Q.W. carried out the SCLC test. R.S. conducts AFM measurement. W.Y.Y. and J.M. wrote the manuscript.

## Competing interests

The authors declare no competing interests.
