## [Peer Review File · Nature Communications]

Reviewers' comments:

Reviewer #1 (Remarks to the Author):

In this work, a non-fullerene acceptor named BTTT-2Cl was synthesized, and a donor/acceptor blend with PZ1 polymer additive was reported with good thermal stability. Considering the novelty and technical issues, I cannot recommend publishing this work in this journal. Please see the detailed comments below:

- (1) The novelty of the new acceptor BTTT-2Cl is limited, the core of the molecule is same as Y6 (ref. 46), but the PCE of PM6/BTTT-2Cl based device is lower than that of PM6/Y6 based device (13.8% vs. 15.5%). What's the advantage of this new molecule compare with Y6?
- (2) What's the original reason of the higher VOC of device with PZ1? The addition amount of PZ1 is only 1%, the enhanced VOC should be discussed.
- (3) The stability of PZ1 based device is very good. The authors claimed the reason for this good stability is "suppressing thermal-induced BTTT-2Cl aggregation in the PM6 polymer domains". In Fig. 3B, this reviewer found PZ1 can reduce the thermal-induced aggregation (crystallization), but even in amorphous OPVs, the thermal stability can not be that good. In my opinion, this is maybe one of the reasons for the good stability, but not the major one, some additional studies are needed.
- (4) The original chemical/physical interactions between PZ1 with PM6/BTTT-2Cl which induced the excellent stability should be the most important part of the work, but missing now.
- (5) The whole part of "space environments, including Moon, Low Earth Orbital Satellite and Mars" should be removed. Since the authors only considered about the temperature change, but missed irradiations in space. The irradiation stability (like X-ray, Gamma-ray) of solar cells is very critical to evaluate the stability in space, that's why some other type cells (like CIGS, GaAs, perovskite) show good potential in space. The authors cannot claim that OPV can be used in space (Moon, Low Earth Orbital Satellite and Mars....) without figuring out the irradiation instability. This part will seriously mislead readers.
- (6) In some previous works, some polymer additives were introduced and employed in OPV for better PCE or stability. What about the unique advantages or new mechanisms of this PZ1 polymer additive?

Reviewer #2 (Remarks to the Author):

This manuscript describes the role of PZ1 as a third component in ternary organic solar cells. The PM6:BTTT-2Cl:PZ1 device achieved a power conversion efficiency of 15.1% with enhanced all photovoltaic parameters compared to control binary device. The authors argued that addition of PZ1 improves the morphology and thermal stability. Overall, the manuscript is well organized and can be accepted for publication in this journal after considering the following issues.

1. Photovoltaic performances of PM6:PZ1 devices are should be given.
2. How about the thermal stability of PM6:PZ1 devices? Based on the contents of manuscript, PM6:PZ1 devices could demonstrate superb thermal stability.
3. Did all thermal stability test be carried out in N₂-filled glovebox or ambient air? It should be notified in the manuscript.
4. In Figure 2C, the authors argued that the lifetime from 0.36us to 0.34us by adding PZ1 indicates less trap-assisted recombination of the PZ1-doped device. However, in Figure 3F, the authors mentioned that small change of lifetime in the PZ1-doped device after 24h thermal stress from 0.34 to 0.36 indicates good thermal stability of the PZ1-doped device. In my opinion, 0.36 and 0.34 within ~20 ns changes in Figure 2C are almost identical values in the error range of measurements like

Figure 3F. It is hard to say less recombination in PZ1-doped device by Figure 2C.

Reviewer #3 (Remarks to the Author):

In this manuscript, the authors have reported that a small amount of PZ1 additive can enhance both the efficiency and stability of the PM6:BTTT-2Cl-based OSCs. The PCE of the device is enhanced from 13.8% to 15.1%. The device exhibit excellent thermal stability after annealing at 150 oC for 800 h. The effect of PZ1 additive on the film morphology, device performance and device thermal stability is thoroughly studied. The excellent thermal stability of the OSC device is very interesting and important. Recently, the PCE of OSCs has exceeded 17%, so the stability of OSCs become one of the most important issue but this issue is rarely solved. This work is a great example to improve the stability of OSCs and the results are amazing. Overall, I think this manuscript is very important and that it should be accepted by Nat. Commun. after revision. The following is some suggestions for revision of the manuscript.

(1) Why a tiny amount (1%) of PZ1 can greatly improve the stability at high temperature. The authors need to give the reason. The thermal stability is the most striking point of this work but the thermal stability is not thoroughly studied and discussed. More study on thermal stability, including the reason, the evolution of AFM image under thermal annealing, the evolution of GIWAXS data, etc..

(2) The thermal behaviors (e.g. DSC curves, phase transition temperature) of the materials themselves need to be provided and discussed. This information will be very important to understand the thermal morphology stability of the OSC devices.

(3) Figure 3b, the AFM images of the active layers under thermal annealing. For the device without PZ1, the morphology does not change obviously from 0 h to 5 h but change dramatically from 5 h to 24 h. More data need to be provided to show the evolution of the morphology. Does the large crystalline domains suddenly appear or grow gradually?

(4) The Figure 3c is not in accordance with Figure S16. In Figure S16, The 001 dot indicated with red arrow in fig. b and fig. c can also be clearly observed in fig. f. In Fig 3c, the diffraction bands cannot be observed in the line cut curves (the purple line at the bottom). The authors need to double check the data.

(5) In the GIWAXS measurements section, the authors claimed that addition of PZ1 results in a favorable blend film with more obvious π - π stacking of BTTT-2Cl molecules. However, the PZ1 doped devices showed the improved hole-mobility and depressed electron-mobility. These results seem contradictory. Please discuss it in the main text.

Responses to the reviewers' comments are as follows:

Response to Reviewer #1:

Comments: In this work, a non-fullerene acceptor named BTTT-2Cl was synthesized, and a donor/acceptor blend with PZ1 polymer additive was reported with good thermal stability. Considering the novelty and technical issues, I cannot recommend publishing this work in this journal. Please see the detailed comments below:

1. The novelty of the new acceptor BTTT-2Cl is limited, the core of the molecule is same as Y6 (ref. 46), but the PCE of PM6/BTTT-2Cl based device is lower than that of PM6/Y6 based device (13.80% vs. 15.5%). What's the advantage of this new molecule compare with Y6?

Response: Thanks very much for the reviewer's comments. Actually, Zou et al. reported the PM6:Y6 system with a PCE of 15.7% in a champion device fabricated under 0.5% chloronaphthalene (CN) as a solvent additive with thermal annealing at 110 °C for 10 min. In our work, applying 1 wt% PZ1 solid additives into the PM6/BTTT-2Cl system, we achieved a PCE of 15.10%, which is very close to the reported value of 15.7% (*Joule*, **2019**, *3*, 1140-1151). As shown in Figure 1, we provided the Y6 derivatives based on the Cl-substitutions and a 3rd-position branched alkyl chain reported by Hou et al. (*Nat. Commun.*, **2019**, *10*, 2515) and Yan et al. (*Joule*, **2019**, *3*, doi.org/10.1016/j.joule.2019.09.010), respectively. It can be easily found that these two Y6 derivatives have the same core of the molecule Y6, even though both Y6 derivatives show the slightly higher PCEs of 16.50% and 16.74%, respectively.

In this work, although the molecule BTTT-2Cl compared with Y6 exhibit a slightly low PCE of 15.10%, its active layer with 1 wt% PZ1 show longer lifetime with nearly keeping their efficiency (t = 800 h) under accelerated heating test at 150 °C. In contrast, as shown in Figure 5, the PCE of Y6-based active layer degraded obviously in the time period of 48 h. Undoubtedly, as compared to the Y6, the BTTT-2Cl-based blends can be significantly used in extremely harsh environments, or even in the outer space applications.

Apart from the highly stable BTTT-2Cl-based blends with the PCEs of over 15% reported in this work, we also reported a micro-doping strategy to improve the thermal stability of photovoltaic systems *via* the use of PZ1 employed as the dual function additive. This general strategy was confirmed by the other four BHJ blends, including PM6:Y6, J71:ITIC, PTB7-Th:PC₇₀BM and BDT-3T-R:PC₇₀BM systems. We believe that this part analysis also highlights the novelty of this article. Meanwhile, we also believe that this strategy can be applied broadly to improve device efficiency and thermal stability.

In this work, our highlights include: (1). A new non-fullerene acceptor, named BTTT-2Cl, achieving a PCE of 15.10%, which is close to that of Y6. (2). A PZ1-doped PM6:BTTT-2Cl photovoltaic system show the excellent thermal stability at high temperature. (3). This BTTT-2Cl-based system subjected to different thermal cycling stress conditions point to a bright future in the outer space applications. (4). A PZ1-doping general strategy has been demonstrated.

Figure 1. The reported articles based on Y6 and its derivatives.

2. What's the original reason of the higher V_{oc} of device with PZ1? The addition amount of PZ1 is only 1%, the enhanced V_{oc} should be discussed.

Response: With 1 wt% PZ1, the device showed the improved V_{oc} values from 0.896V to 0.904V. In fact, the V_{oc} improvement of PZ1-doped device is not obviously, only 0.008V difference. Generally, this slight difference is negligible.

Here I would like to give possible reasons as followed: on the one hand, the higher LUMO energy level of PZ1 can slightly improve V_{oc} value. On the other hand, as discussed in Figure 3, as compared to the PM6:BTTT-2Cl blend, the PZ1-doped active layers showed the better phase separation with slightly larger nanoscale domains. In addition, as shown in Table 1, the higher J_{sc} and FF values of PZ1-doped devices demonstrated that the PZ1-doped blend should have less carrier recombination loss. Thus, the slightly voltage shift ($\Delta V_{oc} = 0.008V$) partially expected from the observed change in charge density or recombination in devices observed in Figure 3E (*Adv. Energy Mater.*, **2015**, *5*, 1500111 and *Energy Environ. Sci.*, **2019**, *12*, 2518-2528).

In the main text, we also provided an explanation marked in red: “Of note is that the slightly voltage shift ($\Delta V_{oc} = 0.008V$) probably resulted from the high LUMO energy level of PZ1 and the less carrier recombination loss, which will be mentioned in below.” And “The reduced recombination order value demonstrated the slightly V_{oc} improvement in in the PZ1-doped device.”.

3. The stability of PZ1 based device is very good. The authors claimed the reason for this good stability is "suppressing thermal-induced BTTT-2Cl aggregation in the PM6 polymer domains". In Fig. 3B, this reviewer found PZ1 can reduce the thermal-induced aggregation (crystallization), but even in amorphous OPVs, the thermal stability can not be that good. In my opinion, this is maybe one of the reasons for the good stability, but not the major one, some additional studies are needed.

Response: Thank you very much for the reviewer's comments. We have reviewed some amorphous photovoltaic systems based on fullerene derivatives as acceptors (*Adv. Funct. Mater.* **2015**, *25*, 748; *Org. Electron.* **2016**, *38*, 15; *ACS Nano* **2011**, *5*, 6233; *J. Phys. Chem. C.* **2018**, *122*, 9843-9851). These systems showed poor thermal stability mainly due to the PCBM phase segregation and aggregation in the blend films. As shown in Figure 5B, the addition of PZ1 can effectively reduce the aggregation of PC₇₁BM and enhance the thermal stability of PBT4-Th:PC₇₁BM system.

In addition, we are in a good agreement with the reviewer's opinion that "suppressing thermal-induced BTTT-2Cl aggregation in the PM6 polymer domains" is one of the reasons for the good stability. Here in the main text we revised the sentence to "*These images provide a direct evidence that PZ1 is very efficient in suppressing thermal-induced BTTT-2Cl migration and aggregation in the blend and also reducing its phase segregation*" and "*Incorporating polymer acceptor PZ1 with long alkyl side chains into the PM6:BTTT-2Cl blend can form an insoluble framework and also inhibit BTTT-2Cl aggregation and crystallization.*". This statement should be more accurate.

It was found that the other reasons can also influence the thermal stability of blends, i.e., molecular weight (*ACS Appl. Mater. Interfaces* **2019**, *11*, 18555-18563), miscibility issue (*Adv. Energy Mater.* **2019**, 1803394), crosslinking of the donor and/or acceptor (*Sol. Energ. Mat. Sol. Cells.* **2019**, *200*, 109982), spinodal donor-acceptor demixing (*Nat. Commun.*, **2017**, *8*, 14541) and interface stratification (*J. Mater. Chem. A*, **2019**, *7*, 23361-23377). However, in this work, it can be easily found that the homogeneity of the PM6:BTTT-2Cl blend was altered subsequent to the heating and numerous long string-like BTTT-2Cl aggregates were observed, whereas the PZ1-incorporated blend morphology was not significantly affected by heating (see Figure 3B and 3C as well as Figure S10). The obvious morphology changes indicate that PZ1 is very efficient in suppressing thermal-induced BTTT-2Cl migration and aggregation in the blend and also reducing its phase segregation. We believe that the other reasons can also result in the poor thermal stability in this system. However, these points are beyond the scope and motivation of this article.

4. The original chemical/physical interactions between PZ1 with PM6/BTTT-2Cl which induced the excellent stability should be the most important part of the work, but missing now.

Response: Thank you very much for the reviewer's comments. For the original chemical interactions between PZ1 with PM6/BTTT-2Cl, we didn't find obvious hand-grabbing-like or crosslinking-like chemical interactions induced between PZ1 and donor PM6 or acceptor BTTT-2Cl materials. On the one hand, their chemical structures of these photovoltaic materials can not give the corresponding information. On the other hand, we only added 1wt% PZ1 into the host blend. The thermal stability of PZ1-doped blends can not be explained by the relevant chemical interactions.

We found that the PZ1 additives can effectively improve the blend with more obvious nanoscale domains and help it to form the interpenetrating network morphology, as shown in Figure 1D. According to GIWAXS, addition of PZ1 results in a favorable blend film with more obvious π - π stacking of BTTT-2Cl molecules (Figure 1D). In addition, the molecular dynamics modeling was used to explain the thermal-driven phase separation and BTTT-2Cl aggregations with and without PZ1 (Figure 3A). The π - π separation of the molecular chains was restrained to model varying levels of confinement, and the resulting dihedral distributions were compared to characterize the BTTT-2Cl reorganization dynamics. This approach is followed to the reported article (*Science*, **2018**, 362, 1131–1134).

We measured the AFM topography images of the PM6:BTTT-2Cl blend as a function of annealing time at 150 °C (see Figure S19 in the SI). The homogeneity of the PM6:BTTT-2Cl blend was altered subsequent to the heating and numerous long string-like BTTT-2Cl aggregates were observed. In addition, we conducted the DSC measurements (Fig. S6), which demonstrated that the crystalline nature of BTTT-2Cl molecules can easily result in their aggregations. However, as shown in Figure 3B, incorporating polymer acceptor PZ1 with long alkyl side chains into the PM6:BTTT-2Cl blend can form an insoluble framework and also inhibit BTTT-2Cl aggregation and crystallization. We added the explanations in the main text: *“Thermal properties of these three materials, including PM6, BTTT-2Cl and PZ1, were investigated by differential scanning calorimetry (DSC). The results are provided in Fig. S6. Only BTTT-2Cl exhibits single, sharp and strong endothermic peak at the first heating (Fig. S6B), which indicates its crystalline nature. As shown in Fig. S6A, PM6 has slightly sharper endothermic peaks at low temperature and with small value of melting enthalpy. It indicates that PM6 possesses slightly higher crystallinity as compared to PZ1 (Fig. S6C). It is in a good agreement with the previous results.^{11, 45,} and “Incorporating polymer acceptor PZ1 with long alkyl side chains into the PM6:BTTT-2Cl blend can form an insoluble framework and also inhibit BTTT-2Cl aggregation and crystallization.”*

5. The whole part of "space environments, including Moon, Low Earth Orbital Satellite and Mars" should be removed. Since the authors only considered about the temperature change, but missed irradiations in space. The irradiation stability (like X-ray, Gamma-ray) of solar cells is very critical to evaluate the stability in space, that's why some other type cells (like CIGS, GaAs, perovskite) show good

potential in space. The authors cannot claim that OPV can be used in space (Moon, Low Earth Orbital Satellite and Mars....) without figuring out the irradiation instability. This part will seriously mislead readers.

Response: Thank you very much for the reviewer's comments. Assuredly, irradiation stability is of vital importance for space application; hence, to promote the development of this frontier field, OPV researchers spend much effort in analyzing and overcoming the irradiation degradation including X-ray (*Adv. Funct. Mater.* **2010**, *20*, 2729; *Energy Environ. Sci.*, **2011**, *4*, 4917) and Gamma-ray irradiation (*J. Phys. D: Appl. Phys.* **2014**, *47*, 015105; *ACS Appl. Mater. Interfaces*, **2019**, *11*, 21741). Particularly, some exciting news in OPV field connected with outer space utilization attempts have been reported during these recent years (<https://www.i-meet.wm.uni-erlangen.de/2018/09/nasa-brought-joses-organic-solar-cells-successful-to-the-outer-space/>; <https://infinitypv.com/applications/other/oscar>). Apart from the irradiation degradation, we considered that temperature issue matters equally in the space application. Thus, we here report a convenient and effective way to universally enhance the OPV's tolerance towards extreme temperatures. Besides, the radiation (ultraviolet ray, X-ray and Gamma-ray) can be filtered out if they really have a big impact on the stability of individual system (*Adv. Mater.* **2018**, *30*, 1800855).

In this main text, we mainly focus on the thermal stability issues and provide a general strategy to improve the thermal stability of photovoltaic systems. Based on this point, we added a sentence to explain the importance of irradiation stability in the hope of reducing the misleading effect on readers: "*Of note is that the thermal stability issues are significantly investigated in this work. Thus, the irradiation stability of relevant active layers was not considered in the simulated environment of outer space applications.*", introduced in the last section.

6. In some previous works, some polymer additives were introduced and employed in OPV for better PCE or stability. What about the unique advantages or new mechanisms of this PZ1 polymer additive?

Response: Thank you very much for the reviewer's comments. Assuredly, improving stability or efficiency by adding polymer additives has been reported previously (*npj Flex. Electron.*, **2017**, *1*, 11, *Energy Environ. Sci.*, **2016**, *9*, 3464-3471; *Nano Energy*, **2019**, *58*, 724-731). However, the test temperatures of these photovoltaic systems are generally in the range of 60 to 120 °C. In fact, their device performance of relevant active layers degraded rapidly at the temperature of 120 °C (*Energy Environ. Sci.*, **2016**, *9*, 3464-3471). Our results show that PZ1 can be employed as the dual function additive (improve device efficiency and thermal stability), and demonstrated by the investigated five photovoltaic systems. It indicates the good generality of this PZ1-doping strategy. In addition, we designed and synthesized a new acceptor BTTT-2Cl. Blending with polymer donor PM6, the PZ1-doped PM6:BTTT-2Cl active layer show the excellent thermal stability at a higher temperature of 150 °C as

compared to the above-mentioned publications. Because of the addition of PZ1, we were able to obtain such a more thermal stable photovoltaic system as compared to the other reports in the literature. Of note is that the PCE of 1 wt% PZ1-doped PM6:BTBT-2Cl system can be improved to 15.10%. Combining with the analysis of thermal stability, PZ1-doped PM6:BTBT-2Cl system is found to be a promising candidate for potential applications of OSCs.

Overall, in this work our highlights include: (1). A new non-fullerene acceptor, named BTBT-2Cl, achieving a PCE of 15.10%, which is close to that of Y6. (2). A PZ1-doped PM6:BTBT-2Cl photovoltaic system show the excellent thermal stability at high temperature (150 °C). (3). This BTBT-2Cl-based system subjected to different thermal cycling stress conditions point to a bright future in the outer space applications. (4). A PZ1-doping general strategy has been demonstrated.

Response to Reviewer #2:

Comments: This manuscript describes the role of PZ1 as a third component in ternary organic solar cells. The PM6:BTBT-2Cl:PZ1 device achieved a power conversion efficiency of 15.1% with enhanced all photovoltaic parameters compared to control binary device. The authors argued that addition of PZ1 improves the morphology and thermal stability. Overall, **the manuscript is well organized and can be accepted for publication in this journal after considering the following issues.**

1. Photovoltaic performances of PM6:PZ1 devices are should be given.

Response: Thank you very much for the reviewer's suggestion. We have optimized the PM6:PZ1 devices. The best efficiency exhibited a PCE of 8.27%. The relevant data are shown in Figure S14 in the SI. In addition, we added the device results in the main text and mentioned "*Besides, the J-V curve of the optimized PM6:PZ1 device with a weight ratio of 1.75:1 is shown in Fig. S14. The best efficiency exhibited a PCE of 8.27% ($V_{oc} = 0.0.94$ V, $J_{sc} = 15.34$ mA cm⁻², FF = 57.65%). It is lower than the published value (11.2%) reported in the Ref. 48, probably resulting from the different molecular weights of PM6 and PZ1.*". Besides, the detailed process of device fabrication provided in the Experimental Section (*Fabrication and characterization of the OSCs*).

Figure 2. J - V curves of PM6:PZ1 (1.75:1, wt%) devices measured under one sun illumination. The average PCE value of 8.02% was calculated by eight devices.

2. How about the thermal stability of PM6:PZ1 devices? Based on the contents of manuscript, PM6:PZ1 devices could demonstrate superb thermal stability.

Response: We have tested the thermal stability of the corresponding PM6:PZ1 blend as a function of heating time at 150 °C. Our results demonstrated that the PM6:PZ1 all-polymer system also possesses the high thermal stability. The relevant figure was exhibited in Figure S18 in the SI. In addition, we added the detailed results in the main text and mentioned “*Unsurprisingly, as shown in Fig. S18, the above-mentioned PM6:PZ1 all-polymer photovoltaic system also retained approximately 90% of its PCEs after 48h, indicating the super thermal stability at a high temperature*”.

Figure 3. Normalized PCE of the PM6:PZ1 blend as a function of heating time at 150 °C.

3. Did all thermal stability test be carried out in N₂-filled glovebox or ambient air? It should be notified in the manuscript.

Response: Thank you very much for the reviewer's suggestion. All thermal stability test were carried out in N₂-filled glovebox in this work. We added the information in the manuscript and marked in red. For instance, "*In parallel, the excellent film quality effectively suppressed thermal-driven phase separation at elevated temperatures, and was thermally stable at 150 oC for over 800 h in the N₂-filled glove box.*" and "*Importantly, we conducted a continuous thermal stress stability test for the PM6:BTBT-2Cl blends without and with 1 wt% PZ1 at 110 °C and 150 °C, respectively, in a nitrogen atmosphere.*", respectively.

4. In Figure 2C, the authors argued that the lifetime from 0.36us to 0.34us by adding PZ1 indicates less trap-assisted recombination of the PZ1-doped device. However, in Figure 3F, the authors mentioned that small change of lifetime in the PZ1-doped device after 24h thermal stress from 0.34 to 0.36 indicates good thermal stability of the PZ1-doped device. In my opinion, 0.36 and 0.34 within ~20 ns changes in Figure 2C are almost identical values in the error range of measurements like Figure 3F. It is hard to say less recombination in PZ1-doped device by Figure 2C.

Response: Thank you very much for the reviewer's comments. We are fully agree with your comments that it is not easy to say less carrier recombination in PZ1-doped device demonstrated by the TPC results. Thus, we deleted the TPC data in Figure 2C, and also removed the discussion of the corresponding results. Please check the modified Fig. 2C in the main text.

Response to Reviewer #3:

Comments: ...The excellent thermal stability of the OSC device is very interesting and important. Recently, the PCE of OSCs has exceeded 17%, so the stability of OSCs become one of the most important issue but this issue is rarely solved. This work is a great example to improve the stability of OSCs and the results are amazing. Overall, **I think this manuscript is very important and that it should be accepted by Nat. Commun. after revision.** The following is some suggestions for revision of the manuscript.

1. Why a tiny amount (1%) of PZ1 can greatly improve the stability at high temperature. The authors need to give the reason. The thermal stability is the most striking point of this work but the thermal stability is not thoroughly studied and

discussed. More study on thermal stability, including the reason, the evolution of AFM image under thermal annealing, the evolution of GIWAXS data, *etc.*

Response: Thank you very much for the reviewer's comments and suggestions. The super thermal stability of PZ1-doped PM6:BTTT-2Cl are mainly contributed to the physical interactions between PZ1 and active layer. During the process of film formation, a tiny amount of PZ1 can facilitate the phase separation, and help host active layer to form the larger interconnected regions and domains as well as the more obvious π - π stacking of BTTT-2Cl molecules as compared to the BTTT-2Cl molecules in the un-doped blend, which demonstrated by the AFM and GIWAXS measurements (Fig. 1D). The molecular dynamics modeling also demonstrates that the π - π confinement exhibited by the obvious phase separation in PZ1-doped blends played a critical role in restricting intrachain reorganization and enabling temperature-insensitive mobility (Fig. 3A). In addition, we believe that the long alkyl side chains of PZ1 with a high molecular weight can effectively prohibit the migration of small molecule acceptor BTTT-2Cl. Thus, we measured the AFM topography images of the PM6:BTTT-2Cl blend as a function of annealing time at 150 °C (see Figure S19 in the SI), and also mentioned in the main text: "*The AFM images of the PM6:BTTT-2Cl blend as a function of annealing time at 150 °C are shown in Fig. S19.*". The homogeneity of the PM6:BTTT-2Cl blend was altered subsequent to the heating and numerous long string-like BTTT-2Cl aggregates were observed. In contrast, the PZ1-doped blend didn't show the obvious morphology evolution. The AFM results are also demonstrated by the GIWAXS data, as shown in Fig. 3C and Fig. S20. Besides, we also provided a detailed explanation in the main text: "*These images provide a direct evidence that PZ1 is very efficient in suppressing thermal-induced BTTT-2Cl migration and aggregation in the blend and also reducing its phase segregation. Incorporating polymer acceptor PZ1 with long alkyl side chains into the PM6:BTTT-2Cl blend can form an insoluble framework and also inhibit BTTT-2Cl aggregation and crystallization.*".

2. The thermal behaviors (e.g. DSC curves, phase transition temperature) of the materials themselves need to be provided and discussed. This information will be very important to understand the thermal morphology stability of the OSC devices.

Response: Thank you very much for the reviewer's suggestions. We measured the thermal behaviors of these three materials investigated by DSC. As shown in Fig.S6, the obvious BTTT-2Cl aggregation and crystallization in the annealed PM6:BTTT-2Cl blend is mainly contributed to its crystalline nature. The amorphous PZ1 can effectively suppress the aggregation behaviors of BTTT-2Cl molecules in the blend. We also provided a detailed expression and marked in red: "*Thermal properties of these three materials, including PM6, BTTT-2Cl and PZ1, were investigated by differential scanning calorimetry (DSC). The results are provided in Fig. S6. Only BTTT-2Cl shows single, sharp and strong endothermic peak at the first heating (Fig.*

S6B), which indicates its crystalline nature. As shown in Fig. S6A, PM6 has slightly sharper endothermic peaks at low temperature and with small value of melting enthalpy. It indicates that PM6 possesses slightly higher crystallinity as compared to PZ1 (Fig. S6C). It is in a good agreement with the previous results.^{11, 45}”.

3. Figure 3b, the AFM images of the active layers under thermal annealing. For the device without PZ1, the morphology does not change obviously from 0 h to 5 h but change dramatically from 5 h to 24 h. More data need to be provided to show the evolution of the morphology. Does the large crystalline domains suddenly appear or grow gradually?

Response: Thank you very much for the reviewer’s suggestions. We re-tested the AFM topography images of PM6:BTTT-2Cl blend films as a function of annealing time, including, 0 h, 1h, 2h, 5h, 8h, 12 h, 16 h and 24h, at 150 °C, as shown in Fig. S19 and also provided in below. It can be easily found that the large crystalline domains grow gradually, which is also in a good agreement with our previous work (*Adv. Energy Mater.* **2016**, *6*, 1502579). Combining with the AFM and GIWAXS investigation of PZ1-doped blend (Fig. 3B, 3C and Fig. S20), our analysis shows that the homogeneity of the PM6:BTTT-2Cl blend was altered subsequent to the heating and numerous long string-like BTTT-2Cl aggregates were observed gradually, whereas the PZ1-incorporated blend morphology was not significantly affected by heating.

Figure 4. AFM topography images of PM6:BTTT-2Cl blend films as a function of annealing time, including, 0 h, 1h, 2h, 5h, 8h, 12 h, 16 h and 24h, at 150 °C.

4. The Figure 3c is not in accordance with Figure S16. In Figure S16, The 001 dot indicated with red arrow in fig. b and fig. c can also be clearly observed in fig. f. In Fig 3c, the diffraction bands cannot be observed in the line cut curves (the purple line at the bottom). The authors need to double check the data.

Response: Thank you very much for the reviewer’s reminding. We double checked

the data. In Figure S20F, the 001 dot can be also clearly observed. The arrows were added in the Fig. 3C and Fig. S20F, respectively. Indeed, we can find a diffraction band in the purple line at the bottom.

Figure 5. The 1D GIWAXS line curves of the corresponding blends with respect to the in-plane direction and out-of-plane direction.

5. In the GIWAXS measurements section, the authors claimed that addition of PZ1 results in a favorable blend film with more obvious π - π stacking of BTTT-2Cl molecules. However, the PZ1 doped devices showed the improved hole-mobility and depressed electron-mobility. These results seem contradictory. Please discuss it in the main text.

Response: Many thanks for the reviewer’s pointing out the results. We have checked the J - V curves and calculated the mobility data again. The average electron mobility ($2.25 \times 10^{-3} \text{ cm}^2 \text{ V}^{-1} \text{ s}^{-1}$) of PZ1-doped blend is still slightly lower than that of the un-doped PM6:BTTT-2Cl blend ($2.07 \times 10^{-3} \text{ cm}^2 \text{ V}^{-1} \text{ s}^{-1}$). Our thought is that these two averages are actually within the margin of error. The error not only resulted from the error of the thickness measurement of active layers for calculating the corresponding mobility, but also are contributed to the device structure with a diode architecture of ITO/ZnO/Active layer/Ca/Al. The ZnO morphology as well as its surface energy can also hugely changed the blend morphology (*Adv. Energy Mater.* **2018**, *8*, 1801807; *Energy Environ. Sci.*, **2019**, *12*, 2518-2528; *Energy Environ. Sci.*, **2015**, *8*, 3442-3476; *Adv. Energy Mater.* **2012**, *2*, 1333–1337; *ACS Appl. Mater. Interfaces*, **2018**, *10*, 12913-12920), which can be not a good in agreement with the AFM and GIWAXS measurements of the corresponding samples based on the PEDOT:PSS-based substrates. In the main text, we also provided a necessary discussion in the main text and marked in red: “*In addition, the average electron mobility values of the relevant devices without and with 1 wt% PZ1 are $2.25 \times 10^{-3} \text{ cm}^2 \text{ V}^{-1} \text{ s}^{-1}$ and $2.07 \times 10^{-3} \text{ cm}^2 \text{ V}^{-1} \text{ s}^{-1}$, respectively. The comparable electron mobility values of the PZ1-doped and un-doped blends are probably contributed to the zinc oxide (ZnO) morphology as well as its lower surface energy as compared to the PEDOT:PSS layer, which affected the film formation of the active layers.*”⁵¹”.

REVIEWERS' COMMENTS:

Reviewer #2 (Remarks to the Author):

The Voc of PM6:PZ1 may be 0.94 V, not 0.0.94 V. It should be revised. Except that one, the manuscript is ready to be accepted.

Reviewer #3 (Remarks to the Author):

I am satisfied with the revision. I think the manuscript can be accepted as is.

Reviewer #4 (Remarks to the Author):

A new organic photovoltaic system with polymeric micro-doping that exhibits outstanding efficiency and thermal stability

This work presents a new acceptor molecule named BTTT-2Cl, and a thermally stable device based on PM6:BTTT-2Cl:PZ1 ternary blend, which retained high efficiency even after 400 hr thermal annealing at 150 °C. The introduction of tiny amounts of PZ1 as a third component greatly improved the long-term thermal stability of the solar cells, which is successfully applied into several different D:A systems. It is also very interesting to find that the authors studied the device stability under rapid temperature change.

The revised manuscript is well organised, and all reviewers' comments have been well and clearly addressed. The presented results are very important and I suggest acceptance of the manuscript after minor revision, after addressing the issues below:

1. Line 122 to 126 and Fig. S6: The authors state " BTTT-2Cl shows single, sharp and strong endothermic peak at the first heating (Fig. S6B)". But it is only possible to see an exothermic peak in Fig. S6B? Cold crystallisation is normally an exothermic process.
2. Line 221: "The values in bracket are the average PCE obtained from eight devices." There are no brackets. This footnote should be deleted.
3. Line 269: "numerous long string-like BTTT-2Cl aggregates were observed" These are not aggregates, they are crystals. According to IUPAC 2013 the definition of an aggregate is: "Irregular cluster of otherwise individual molecules or particles." The features are clearly crystalline according to Fig. 3C. On line 276 it should be "thermally induced crystallisation" and not "thermally induced aggregation".
4. Line 273 to 275: The statement "Incorporating polymer acceptor PZ1 with long alkyl side chains into the PM6:BTTT-2Cl blend can form an insoluble framework" is not clear. What is the polymer not soluble in? Remove insoluble.
5. Line 391 to 395: Device performance of four different systems without using PZ1 before and after heating are compared, but the figures and tables listed in the parentheses and presented in the supporting information actually show the photovoltaic parameters of devices with different PZ1 content, not after heating.

Supporting information

6. Line 57 to 58: It was mentioned that the four different blends were “bladed” on top of PEDOT:PSS. But the active layers were all spin-coated, the use of “bladed” is misleading. I suggest changing the word to “deposited” or “spin-coated”.
7. Line 181: “Fig. S8” should be corrected to “Fig. S9”.

Response to Reviewer #2:

Comment:

The Voc of PM6:PZ1 may be 0.94 V, not 0.0.94 V. It should be revised. Except that one, the manuscript is ready to be accepted.

Response: Thank you very much for the reviewer's comments. We corrected it.

Response to Reviewer #3:

Comment:

I am satisfied with the revision. I think the manuscript can be accepted as is.

Response: Many thanks for the reviewer's kind help and contributions in this article.

Response to Reviewer #4:

Comment: The revised manuscript is well organized, and all reviewers' comments have been well and clearly addressed. The presented results are very important and I suggest acceptance of the manuscript after minor revision, after addressing the issues below:

1. Line 122 to 126 and Fig. S6: The authors state " BTTT-2Cl shows single, sharp and strong endothermic peak at the first heating (Fig. S6B)". But it is only possible to see an exothermic peak in Fig. S6B? Cold crystallisation is normally an exothermic process.

Response: Thank you very much for the reviewer's comments. We corrected the Supplementary Figure 6 in the Supplementary Information file.

2. Line 221: "The values in bracket are the average PCE obtained from eight devices." There are no brackets. This footnote should be deleted.

Response: Thank you very much for the reviewer's suggestion. We deleted the footnote.

3. Line 269: "numerous long string-like BTTT-2Cl aggregates were observed" These

are not aggregates, they are crystals. According to IUPAC 2013 the definition of an aggregate is: “Irregular cluster of otherwise individual molecules or particles.”

The features are clearly crystalline according to Fig. 3C. On line 276 it should be “thermally induced crystallisation” and not “thermal-induced aggregation”.

Response: Thank you very much for the reviewer’s comments. We corrected them in the main text.

4. Line 273 to 275: The statement “Incorporating polymer acceptor PZ1 with long alkyl side chains into the PM6:BTBT-2Cl blend can form an insoluble framework” is not clear. What is the polymer not soluble in? Remove insoluble.

Response: Thank you very much for the reviewer’s suggestion. We removed ‘insoluble’, and changed to ‘...can form a strong framework...’.

5. Line 391 to 395: Device performance of four different systems without using PZ1 before and after heating are compared, but the figures and tables listed in the parentheses and presented in the supporting information actually show the photovoltaic parameters of devices with different PZ1 content, not after heating.

Response: Thank you very much for the reviewer’s comments. We have added the relevant photovoltaic performance and parameters, as shown in Supplementary Figure 24 to 35 and summarized in Supplementary Table 7 to 18 in the Supplementary Information file.

Supporting information

6. Line 57 to 58: It was mentioned that the four different blends were “bladed” on top of PEDOT:PSS. But the active layers were all spin-coated, the use of “bladed” is misleading. I suggest changing the word to “deposited” or “spin-coated”.

Response: Many thanks for pointing out this mistake. We changed to ‘spin-coated’ in the Methods Section.

7. Line 181: “Fig. S8” should be corrected to “Fig. S9”.

Response: We corrected it.